# Technical Note: Assessment of a Novel Method to Measure Water Intake in Beef Cattle and Its Application to Determining Dry Matter Intake

**DOI:** 10.3390/ani15192904

**Published:** 2025-10-04

**Authors:** Hartley J. VanGilder, Nathan E. Blake, Tylor J. Yost, K. E. ArunKumar, Matthew Walker, Ida Holásková, Jarred W. Yates, Matthew E. Wilson

**Affiliations:** 1School of Agriculture and Food Systems, Davis College of Agriculture and Natural Resources, Division of Land Grant Engagement, West Virginia University, Morgantown, WV 26506, USA; hjv00002@mix.wvu.edu (H.J.V.); neb00001@mix.wvu.edu (N.E.B.); tjy0008@mix.wvu.edu (T.J.Y.); eswararunkumar.kalaga@mail.wvu.edu (K.E.A.); 2West Virginia Agricultural and Forestry Experiment Station, Morgantown, WV 26506, USA; mwalke18@mix.wvu.edu (M.W.); ida.holaskova@mail.wvu.edu (I.H.); jerry.yates@mail.wvu.edu (J.W.Y.); 3Office of Statistics and Data Analytics, Davis College of Agriculture and Natural Resources, West Virginia University, Morgantown, WV 26506, USA

**Keywords:** beef cattle, water intake, feed intake, machine learning

## Abstract

**Simple Summary:**

Quantifying individual water intake of beef cattle is important as attention continues to rapidly grow about the importance of water quantity and quality. Measuring individual animal intake is critical to determining the use of this vital resource and to empowering research that benefits from knowing the individual intakes in machine learning and deep learning approaches that depend on “Internet of Things” sensors to characterize animal phenotypes. Here, we assess a new water intake monitoring tool and provide a use case for it in a machine learning approach to leverage the passively collected data.

**Abstract:**

Improving the efficiency, economic viability, and environmental sustainability of beef cattle production requires tools to identify resource-efficient animals. Validated tools to measure, monitor, and verify individual feed and water intake are needed. Here, we verify the validity of the Vytelle In-Pen Weighing Position to passively collect daily full body weights and assess the use of an integrated flow meter with a commercial waterer as a tool to measure daily water intake. This study involved 103 bulls (40 Charolais and 63 Angus) and 54 heifers (25 Charolais and 29 Angus). These animals were fed in a facility with feed intake nodes, In-Pen Weighing, and metered waterers. Body weights collected on the chute scale and total water usage measured by a residential water meter were used to evaluate body weight and water intake measured at the In-Pen Weighing Positions. We confirmed that In-Pen Weighing is highly correlated to chute weighing (Spearman’s correlation coefficient, ρ = 0.99, *p* < 0.0001). We observed high correlation of total water use measured at the In-Pen Weighing units with the facility’s total water use (ρ = 0.9999, *p* < 0.0001). This validates the accuracy of the in-pen water meters, but not the precision of applying water consumption to individual animals. The use of such passive monitoring equipment has the potential to help improve the sustainability of animal agriculture.

## 1. Introduction

Improving our measurements of individual animal performance is key to improving the efficiency of production [1,2,3]. Over the span of 1991 to 2021 the U.S. beef industry reduced its total water footprint by 37%, largely by improving feed efficiency as ~93% of the total water footprint is water used to irrigate plants used as feed [1]. As water scarcity continues to increase, measurement of individual animal water use, and the potential to select for more water use-efficient animals are becoming critical and our assessment of water use in beef cattle is based largely on a paper from 1956 [4] though recent work has started to update those estimates and indicate that water use efficiency is heritable [5,6,7].

Animal agriculture increasingly benefits from advanced technologies and sensors that monitor and measure animal performance (e.g., Vytelle/GrowSafe, C-Lock, smart ear tags, etc. [8,9].) These technologies generate raw data for direct use or for use as variables in much larger datasets, empowering machine learning and data science to produce aggregated animal performance data [3]. These technologies quantify key animal performance data, such as body weight and water intake, without requiring the removal of animals from their production environment (e.g., use of metabolism crates) or costly manual data collection. By collecting data passively in the normal production environment, these technologies help avoid disrupting animal performance or altering behavior through relocation or restrictive housing. Validation is key to ensuring these technologies work effectively in the field. Previously, others have described the validation of Vytelle In-Pen Weighing as a tool to measure body weight passively [10,11]. Validation of one of the existing water intake systems (Insentec) has been described [12,13]. In that work, the authors simply validated that the visual output of the water trough apparatus matched the data logged, not actually validating water use or water consumption. Here, we confirm the findings of others [10,11] for Vytelle In-Pen weighing and provide an external assessment of an integrated custom water flow meter in-line with the commercial water bowl used to measure water intake [3,14].

## 2. Materials and Methods

### 2.1. Animal Breeds and Sourcing

Animals used for this study were housed at West Virginia University Reymann Memorial Farm (Wardensville, WV, USA). The animals were part of a purebred performance test that included Charolais bulls, Charolais heifers, Angus bulls, and Angus heifers. The animals were all private consignments to the performance test, originating from 12 different farms in the Mid-Atlantic region. Upon arrival, the bulls were treated with Safe-Guard^®^ (Merck Animal Health, Madison, NJ, USA), a de-wormer, an intranasal dose of pasteurella vaccine (Bovilis Once PMH, Merck Animal Health, Omaha, NE, USA), Clean-UpTM II (Elanco Animal Health, Greenfield, IN, USA), chute weighed, and ear tagged as per Wardensville Summer Test guidelines. On day 0, the mean weight of the Charolais bulls (*n* =40) was 421 ± 11 kg, Angus bulls (*n* = 63) was 318 ± 6 kg, Charolais heifers (*n* = 25) was 339 ± 10 kg, and Angus heifers (*n* = 29) was 275 ± 10 kg. The average age of the animals upon arrival was 291 ± 2 days.

### 2.2. Collection of Feed Intake Data

Feed Intake Nodes (GrowSafe 8000, Vytelle, Lenexa, KS, USA) were used to collect feed intake data during the tests. Feed Intake Nodes collect feed data via a feed bunk on load cells. To access the feed bunk, an animal must pass its head through a gate with an antenna in the lip of the bunk that reads an animal’s radio frequency identification (RFID) tag [3]. The gate also limits access to one animal at a time. In-Pen Weighing Positions (GrowSafe Beef, Vytelle, Lenexa, KS, USA) were used to weigh animals daily [10]. The In-Pen Weighing Positions consist of a weighing platform positioned in front of the water trough, designed to collect partial body weights paired with an antenna to read the animal’s RFID tag. Each time an animal visits the water trough, its unique RFID number and partial body weight are recorded. Full body weights, used as the Body Weight variable, are calculated via imputation from the front-end weight collected when the animal is in the In-Pen Weighing Position with a resolution of 30 g. The accuracy of the full body weight data based on In-Pen Weighing Position-collected front-end weight has been previously validated [10,11] and we confirm here its validity in our system. Water fed via Ritchie Ecofount waterers was used as an attractant. Each waterer was equipped with a custom flow meter (JLC International, Inc., New Britain, PA, USA) to collect water intake data during the test duration. Water was continuously available as cattle accessed the In-Pen Weighing Positions. The study occurred within a drylot facility containing five pens measuring 29 m × 51 m for the four bull pens (~26 bulls per pen), 36 m × 51 m for the heifer pen (54 heifers) and each including six Feed Intake Nodes and two In-Pen Weighing Positions [3].

For the performance evaluation, animals were on test for 60 days (6 July 2023–4 September 2023), following a 14-day diet acclimation period. Bulls were fed a total mixed ration ad libitum. The diet consisted of 65.6% corn silage, 14.9% mixed grass hay (2.5–5 cm particle length), 14.5% cracked corn, 4.7% supplement (dried distillers’ grains, soybean meal, and wheat middlings), and a commercial vitamin and mineral mix containing selenium. Rumensin^®^ and Tylan^®^ were added at the labeled dosage rate. The ration contained 11.6% crude protein (CP) and 69.9% total digestible nutrients on dry matter (DM = 48.3%) basis. The calculated net energy for maintenance (NEm) and net energy for gain (NEg) were 1.74 and 1.12 Mcal/kg, respectively. Heifers were fed a total mixed ration ad libitum. The diet consisted of 44.3% triticale silage, 37.6% corn silage, 6.6% mixed grass hay (2.5–5 cm particle length) and 4.7% supplement including a commercial vitamin and mineral mix containing selenium. Rumensin^®^ and Tylan^®^ were added at the labeled dosage rate. The ration contained 11.7% CP on a DM = 38.9% basis. The calculated NEm and NEg were 1.65 and 1.06 Mcal/kg, respectively. Ration samples were collected into quart bags from the feed truck after mixing and prior to dispensing into the feed bunks as per Cumberland Valley Analytical sample submission guidelines. Collected samples were refrigerated and overnight shipped to Cumberland Valley Analytical Services (Waynesboro, PA, USA). Nutrient content of the ration was analyzed via Near Infrared Reflectance Spectroscopy. Supplementary white salt was provided ad libitum in each pen during the data collection window. Bulls were weighed using a conventional livestock scale periodically while on the test.

### 2.3. Collection of Water Intake Data

A single pipe, dedicated to the barn, supplied water to the Ritchie waterers. The bowl diameter of the Ritchie Ecofount (Conrad, IA, USA) is only 22 cm and the drink well extends 20 cm down from the top of the waterer, significantly limiting the ability of animals to splash water out of the bowl. Immediately outside the footprint of the barn, a residential water meter was installed to measure total water delivered to the animal pens. A single frost-free hydrant in the barn had a flow meter attached, allowing its use to be quantified and added to the total water measured by the flow meters at the In-Pen Weighing Position water stations. The flow meter data was recorded via the Vytelle Data Acquisition Panel (version 16.63.0.124) and thereby integrated with the RFID recordings and load cell recordings at the In-Pen Weighing Position. The software for the intake system measures all the flow through the meters and then associates the given flows to the RFID tag present in the In-Pen Weighing Position at that time. Proprietary components of the software assign flow (including immediately after an animal leaves the In-Pen Weighing Position) to the same animal to which the weight data is assigned. A dedicated RFID tag was placed adjacent to the antenna in the In-Pen Weighing Positions when water bowls were manually emptied and cleaned so that water measured at the In-Pen Weighing Position that was not consumed by an animal could also be determined and accounted for. For this work, we used the audit function of the Vytelle software to determine the total flow measured by all 10 flow meters in the barn and added the measured hydrant water use to that total. Those flows were added together daily and compared to the accumulated metered volume from the barn meter.

### 2.4. Climate Data

Climate data were downloaded from the Winchester Airport (OKV) NOAA station and included maximum, minimum, mean, and the range of temperatures, humidities, and wind speeds. The data also included daily measures of barometric pressure, precipitation, and solar radiation. The data were joined by their common field, “date”, such that each animal on test for a given date was assigned the climate values collected for that date.

### 2.5. Data Processing

Missing values were imputed using methods tailored to the length of consecutive gaps in each numeric feature. Initially, the dataset was grouped by EID, and imputation techniques were utilized to impute the data specific to each EID. This allowed us to impute the missing values with a random value relative to the actual data distribution. When consecutive missing value gaps were of three or fewer days, the missing values were imputed with the K-Nearest Neighbors [15]. When missing values were edge cases (beginning and ending of the time series features), missing values were imputed with forward-fill and backward-fill methods. When consecutive missing value gaps were more than three days, the missing values were imputed using interpolation. Linear interpolation was applied to estimate missing values where feature trends were temporally steady. This technique was used for imputing independent features, such as age, where more linear change is expected in the distribution over time. For body weight, DMI, daily water intake, daily water visits, time at water, and time per water visit, Piecewise Cubic Hermite Interpolating Polynomial estimates were used [16]. This technique was used due to the nonlinear trends of the intake variables resulting from the feeding behavior of the animals. In our dataset of 6218 records, missing values were limited (dailyWaterVisits (201 records), timeAtWater (210 records), timePerWaterVisit (210 records), dailyWaterIntake (210 records) 3.4% each and bodyweight (52 records) 0.8%). Missingness was imputed using a stepwise approach: long gaps (>3 days) were interpolated (PCHIP for intake and water traits, linear for age), short gaps (≤3 days) were imputed using KNN (k = 6), and edge cases were handled with forward- or backward-fill. This ensured that imputations preserved the temporal structure within the animals rather than relying on overall dataset distributions, thereby minimizing potential bias.

### 2.6. Data Analyses

Body weight data preparation involved exclusion of data with a Vytelle-indicated error in the measurement. The final dataset for comparing the chute versus In-Pen Weighing Position body weight had 147 animals, including 37 Charolais bulls, 59 Angus bulls, 24 Charolais heifers, and 27 Angus heifers. The individual full BW determined from the In-Pen Weighing Positions were compared to chute weights on the days that chute weights were recorded, in a doubly repeated measures approach (animal RFIDs were repeated across days and weight measurements were repeated across weighing methods within a day) [17]. The BW was Ln-transformed to address the lack of normality (determined by the Shapiro–Wilk W test) exhibited by right skewness. The mixed model (SAS software, v.9.4), included the fixed effects of weighing methods (Chute, In-Pen Weighing Position), animal class (bulls, heifers), breed (Angus, Charolais), and time (14 July 2023 and 7 August 2023), with weighing method and time being also used as repeated factors, with unstructured and first order autoregressive covariance structure, respectively [18]. The subject was the individual animal ID for repeated factors, and the breed for random intercepts. The goal was to see if the overall main effect of the method is significant when controlling for breed, animal class, time, and the two-, three-, and four-way interactions. In addition to the doubly repeated model, the nonparametric Spearman correlation of chute and In-Pen Weighing Position measuring methods was calculated together (pooled) and individually (stratified correlation) for each of the eight subgroups, representing the combinations of the two breeds, the two animal classes, and the two time points. To address the possible relatedness of animals within the subgroups (from the statistical design perspective), correlation of the means of the eight groups was assessed by Pearson’s correlation. This enabled us to determine both the “within group” and “between groups” correlations. Lastly, the bias and agreement limits of both weighing methods were determined by Bland–Altman analysis.

Spearman Rho (ρ) correlation of daily water consumption spanning 61 days from 6 July 2023 to 4 September 2023 in the form of cumulative water volume (kL) was calculated between the barn meter and the In-Pen Weighing Position flow meters. In addition, the parallelism of the water usage between the two methods with respect to time was assessed by the parallelism F-test.

In addition to using the feed intake bunks, dry matter intake was determined for the bulls and heifers using a previously developed machine learning algorithm [3,19]. Missing data and imputation were performed. For this report, we used the GPBoost algorithm which is optimized for predicting DMI as reported by ArunKumar [19] in which several modeling approaches were compared using R^2^ and RMSE [14]. Residual feed intake (RFI) was calculated by regressing dry matter intake on metabolic mid-test body weight and average daily gain [20]. Residual water intake was calculated by regressing water intake on metabolic mid-test body weight and dry matter intake [5].

JMP and SAS software (JMP^®^, Version Pro 16.0.0, SAS Institute Inc., Cary, NC, USA, Copyright ©2021; SAS^®^, Version 9.4, SAS Institute Inc., Cary, NC, USA, Copyright ©2002–2012) were utilized for the Shapiro–Wilk W test, Spearman’s correlation analysis, and mixed-model ANOVA. The significance criterion alpha for all tests was 0.05.

## 3. Results

The chute weights and In-Pen Weighing Position weights for all animals, for the days that chute weights were recorded, appear qualitatively identical, depicted by the frequency histogram (Figure 1). All four distributions were lognormal as ranked by Akaike Information Criterion. When analyzed by doubly repeated measures ANOVA, controlling for the breed, animal class, the repeated date, and the interactions, the methods were not different (F = 0.42, *p* = 0.52, Figure 2). The overall pooled Spearman’s correlation between chute and In-Pen Weighing Position weights across all animals and days, was strongly positive (ρ = 0.99, *p* < 0.0001). The breakdown of relationships for all eight subgroups for stratified Spearman’s correlation is in Table 1. In addition, Pearson’s correlation of the means of the eight subgroups was detected (r = 0.95 for all groups, *p* < 0.0001). Of the total of 294 observations, when both chute and In-Pen Weighing Position weights were recorded on the same two days, 14 were outside of the 95% agreement limits determined by Bland–Altman analysis (5%, Figure 3).

When the total barn water use (as measured by the residential meter) was compared to the total flow measured at the ten In-Pen Weighing Positions plus the occasional use at the frost-free hydrant, the Spearman Rho correlation was strongly positive, ρ = 0.9999, *p* < 0.0001 (Figure 4). There was parallelism between the lines of best fit for both the cumulative In-Pen Weighing Position volume and the total barn water, based on the Parallelism F-test (F = 0.135, *p* = 0.714, Figure 5).

On day 60, the mean weight of Charolais bulls (n = 35) was 538 ± 12 kg, Angus bulls (n = 57) was 422 ± 7 kg, Charolais heifers (n = 24) was 396 ± 10 kg, and Angus heifers (n = 28) was 344 ± 9 kg. This reflects the ADG of Charolais bulls of 1.68 ± 0.04 kg/day, Angus bulls of 1.53 ± 0.03 kg/day, Charolais heifers of 0.94 ± 0.05 kg/day, and Angus heifers of 1.04 ± 0.03 kg/day.

The measures of feed and water intake are reported in Table 2. Overall, the measured and determined DMI of animals across all days had an R^2^ ranging from 0.47 to 0.50. The relationship for just Charolais bulls was an R^2^ of 0.50 and RMSE of 1.23 kg, Angus bulls was an R^2^ of 0.50 and RMSE of 1.36 kg, Charolais heifers was an R^2^ of 0.44 and RMSE of 1.33 kg, and Angus heifers was an R^2^ of 0.46 and RMSE of 1.54 kg (Figure 6); Charolais bulls had an average measured feed/gain of 6.17 ± 0.37, an average determined feed/gain of 6.12 ± 0.35, and an average water/gain of 27.6 ± 0.8. Angus bulls had an average measured feed/gain of 5.99 ± 0.29, an average determined feed/gain of 6.03 ± 0.28, and an average water/gain of 24.26 ± 0.7. Charolais heifers had an average measured feed/gain of 11.33 ± 0.46, an average determined feed/gain of 11.33 ± 0.45, and an average water/gain of 35.07 ± 1.0. Angus heifers had an average measured feed/gain of 9.42 ± 0.43, an average determined feed/gain of 9.34 ± 0.42, and an average water/gain of 24.64 ± 1.0.

The resource use efficiency for all animals is depicted in Figure 7. Each animal group has individual animals in all four quadrants, showing that the study includes animals that are efficient in both feed and water use, inefficient in both, or efficient in one while inefficient in the other. For all animals, those with positive and negative RFI (Figure 8) and positive and negative RWI (Figure 9) exist throughout the feed or water intake spectrum.

## 4. Discussion

As concerns continue to grow about the abundance of and sources used to provide water to livestock, improving water use efficiency is important. Klopatek and Oltjen [1] provided an update of the total water footprint of the U.S. Beef Industry and noted that there has been a 37% reduction in the water footprint per unit of product, largely because of improvements in feed efficiency in the industry. Their analysis relies on book values for consumed water, and thus, may overestimate water consumption as that approach is largely based on data from the 1950s [4,21]. Clearly, the ability to measure and monitor individual water consumption in a passive system has great potential to allow the industry to further reduce its water and ecological footprint. 

Technology to enhance the management of animal agriculture has continued to increase in recent years [22]. These technologies could enhance livestock management efficiency [23] and support measuring, monitoring, validating, and verifying practices to reduce animal agriculture’s ecological footprint [2]. Here, we demonstrate that our method of passive weight and water intake monitoring is accurate and therefore has the potential to improve animal management and address questions of intakes and performance without the need for laborious tasks that take animals out of their normal production environment.

Previously, Wells et al. [10] demonstrated very high levels of agreement between traditional chute scales and both walk-over-weighing and In-Pen Weighing Position, indicating the adequacy of passive body weight measurements. An added benefit of these passive weighing technologies is their increased accuracy. When an attractant (e.g., water, mineral) is used, animals weigh themselves multiple times daily. This is likely to provide greater accuracy than a single chute weight, which depends on user judgment while the scale fluctuates as the animal moves. By taking regressed weights based on multiple visits, over extended periods, without the stress of the handling system, automated, passive weight monitoring is likely to be much more accurate [9,10].

As the first deployment of this prototype water meter and algorithm for passive individual water intake [3,19], our system required assessment. Fortunately, our installation occurred in a barn in which the water line feeding the barn was a terminal line and therefore installation of a residential meter for comparison to the measured water use in the barn was possible. Validation of one of the water intake monitoring approaches was suggested to be a valid measurement tool [12,13]. In that work, the visual water readout from the water trough apparatus was compared to the data logged in the computer system. The agreement of this comparison was assumed to validate the water consumed. This approach, however, simply verified the logging of data, with no reported external verification or calibration of the sensors. Here, we compared the sum of all of the consumed water in the barn (as measured by flow meters at the water bowl) to an external measure of total water delivered to the barn, thereby showing agreement between two different measures of the same variable. Feed intake from bunks is assigned by determining a difference in bunk weight and assigning that to the animal that is present. There is no way to assess, in that case, that the feed was consumed. Using the same data collection framework, here, the flow data is integrated into the data acquisition panel and is assigned to the animal that is present (including the flow to complete the bowl recharge immediately after an animal leaves). The flow monitoring is fully integrated into the Vytelle system, the data on individual water intakes are retrieved from the same database where the body weight data are retrieved. The sum of those individual daily water intakes was then compared to the sum of all flows into the facility using a separate meter and a very high level of agreement was found over 60 days (157 animals × 60 days × an average of 5 visits/day ~ 47,000 intake records). This approach to measuring individual water intake appears to be accurate and opens the possibility to conduct a wide range of valuable experiments. These include genetic evaluation of water use and water use efficiency [5,7], assessments of individual water intake to integrate modern genetic data into resources like the Nutrient Requirements of Beef Cattle [21], and evaluations of how different feedstuffs affect water use efficiency.

With the ability to measure daily water intake, it is possible to monitor biological water use efficiency [5,6,7]. Determining the accuracy of such measures is important as these efforts move forward. Recent reports have estimated the heritability of residual water intake as a measure of water use efficiency between 0.39 [5] and 0.45 [7]. Traditional measures of dry matter intake incorporate five of the six nutrient classes (carbohydrate, protein, lipid, vitamin and mineral), and the addition of the sixth nutrient class (i.e., water) allows for an assessment of an animal’s total “fuel efficiency”. It is noteworthy that efficient animals occur throughout the ranges of feed or water intake, leading us to suggest that animals can be both efficient and inefficient at various body weights and levels of gain, reinforcing the need to measure the trait in systems that allow for quantification of individual feed and water intake.

## 5. Conclusions

Animal agriculture needs to continue to become more efficient. Historically, the driver for that was to reduce the input costs of production. In addition to reducing input costs, there is a need to reduce the ecological footprint associated with animal agriculture to achieve greater industry sustainability and allow producers to capture the value of both inset and offset credits. To accomplish greater efficiencies, we need technologies that allow producers to measure individual animal performance. Here, we validate a new approach to measuring individual animal water intake and its use to measure both water and feed efficiency.

Recently, great progress has been made in determining individual feed intake using daily measurement of water intake, body weight, climatic variables and animal metadata [3,16]. Our results show that this technology closely aligns with actual dry matter intake, tracks individual weight and water intake, and could eventually replace bunk-based intake systems for performance assessment and genetic selection of more efficient animals. Methods like those described here have potential applications for quantifying and comparing the intakes of animals in extensive grazing systems, potentially empowering research in the settings where the majority of ruminant animals are found.

## Figures and Tables

**Figure 1 animals-15-02904-f001:**
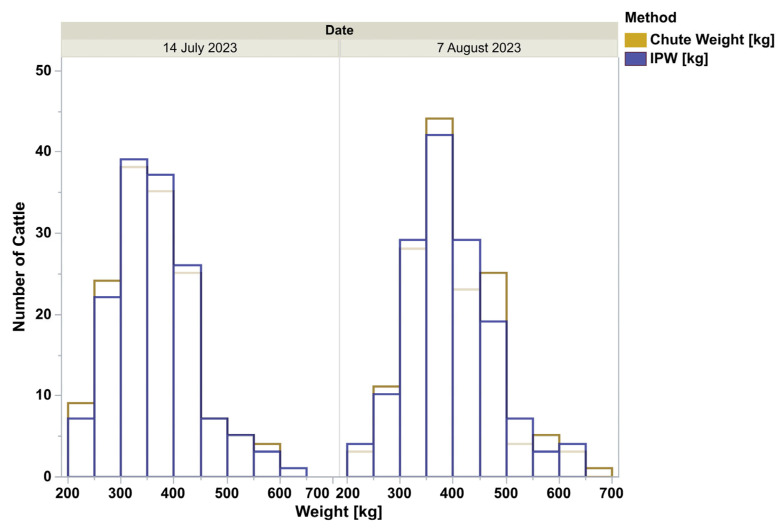
Histograms of body weights collected on 14 July 2023 or 7 August 2023 using either the chute scale (Chute) or the Vytelle In-Pen Weighing (IPW) Positions.

**Figure 2 animals-15-02904-f002:**
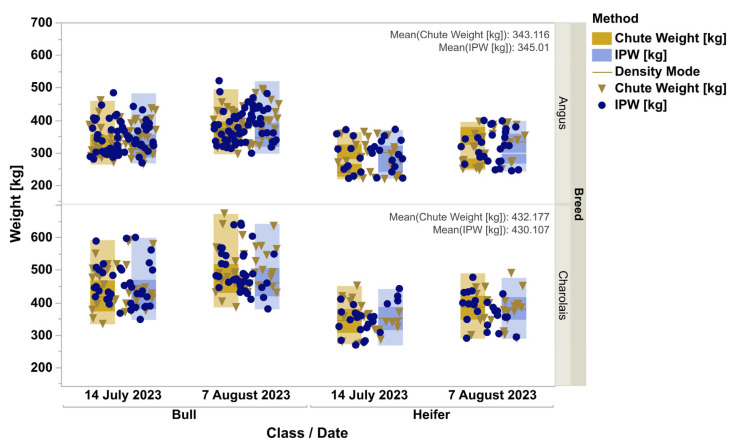
Distribution of chute (gold) and IPW (blue) weights (kg) in Angus and Charolais bulls and heifers on two occasions. The dots represent actual weights of individual animals, and the boxes contour the spread of weighing method data. Within each method (color), the dark region marks 50% and the whole box represents 99% probability density. The horizontal line in each box is the mode. Noticeable are the similarities in the high-density regions between the weighing methods. Considering the impact of breeds, class, and growth in repeated measures ANOVA on body weight, there was no main effect of weighing method detected (*p* = 0.52).

**Figure 3 animals-15-02904-f003:**
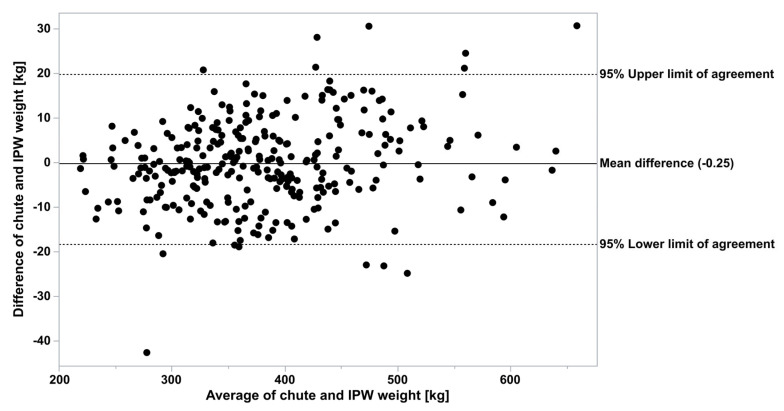
Bland–Altman plot of body weights measured with the chute scale or the Vytelle In-Pen Weighing (IPW) Positions.

**Figure 4 animals-15-02904-f004:**
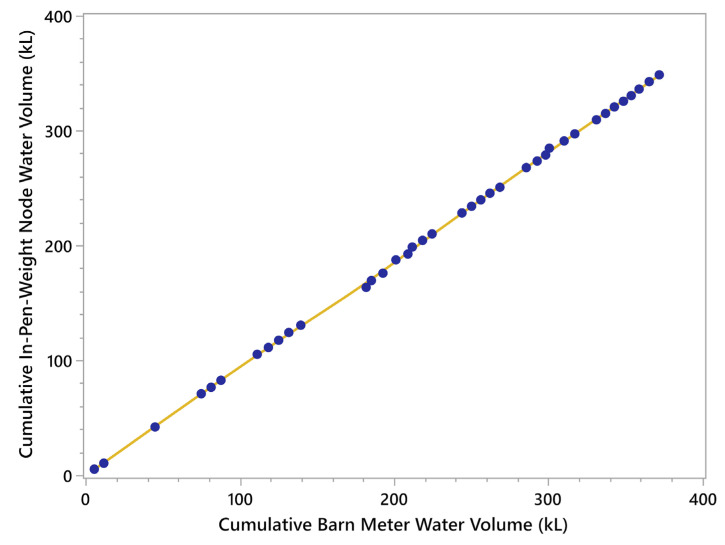
Relationship between the cumulative water consumption measured using the flow meters associated with the In-Pen Weighing Positions and the cumulative water consumption measured using the residential water meter for the barn. Yellow line is the sum of the daily animal consumption and the blue dots are the total volume of water delivered to the facility.

**Figure 5 animals-15-02904-f005:**
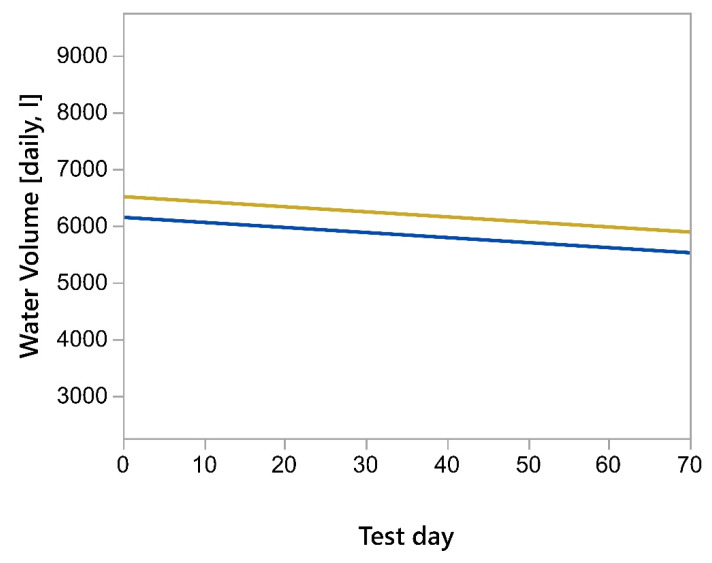
Parallelism plot for the water volume measured by the In-Pen Weighing Position-associated water meters (blue) and the total water volume measured by the barn water meter (gold).

**Figure 6 animals-15-02904-f006:**
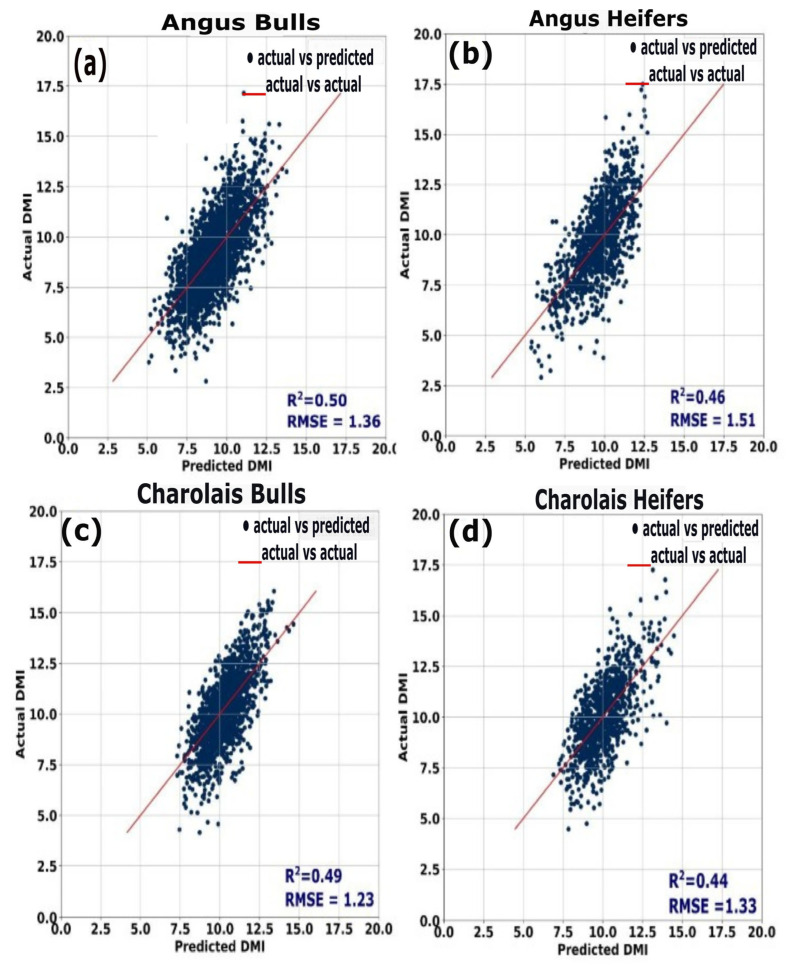
Determined vs. measured dry matter intake (kg) for each breed × class group. Dry matter intake was either determined by the GPBoost method of [19] or measured by the Vytelle feed bunks.

**Figure 7 animals-15-02904-f007:**
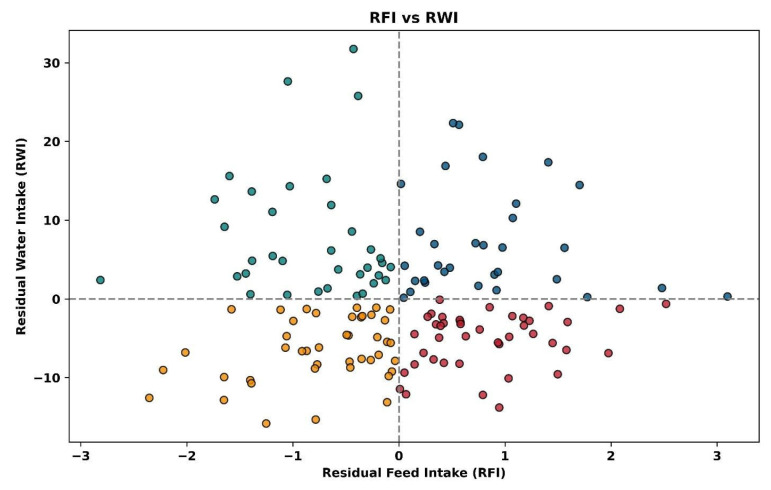
Plot of all individual animals’ residual feed intake (kg) vs. residual water intake (L). Gold dots represent animals efficient in water and feed use, blue dots represent animals inefficient in water and feed use, green dots represent animals inefficient in water use and efficient in feed use and red dots represent animals efficient in water use and inefficient in feed use.

**Figure 8 animals-15-02904-f008:**
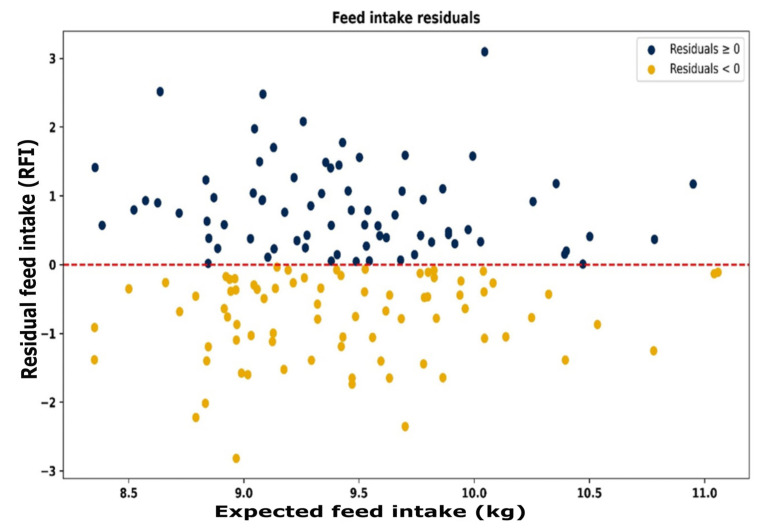
Residual vs. expected feed intake (kg), demonstrating that animals are efficient (positive RFI) or inefficient (negative RFI) throughout the range of expected feed intakes. Red line represents the average efficiency for the group.

**Figure 9 animals-15-02904-f009:**
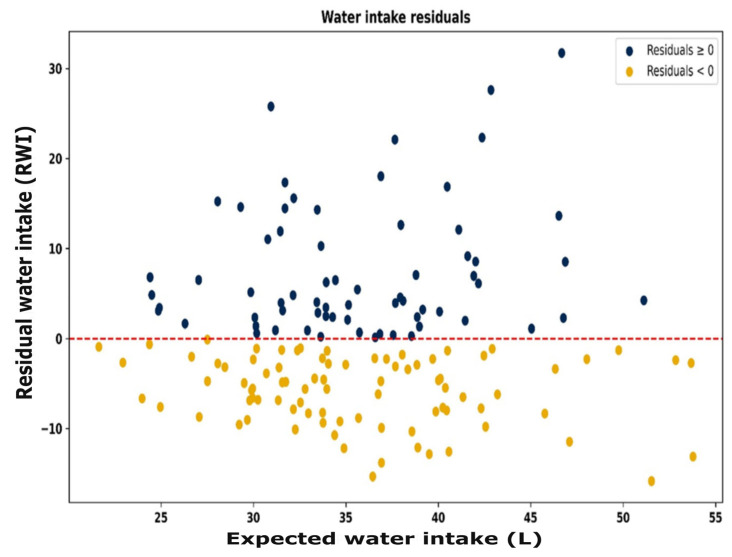
Residual vs. expected water intake (L), demonstrating that animals are efficient (positive RWI) or inefficient (negative RWI) throughout the range of expected water intakes. Red line represents the average efficiency for the group.

**Table 1 animals-15-02904-t001:** Spearman correlation coefficients between chute weights and In-Pen Weighing Position weights for each breed × class group on each day that chute weights were recorded.

Date	Breed	Class	Spearman’s ρ	Prob > |ρ|
14 July 2023	Charolais	Bull	0.9885	<0.0001
7 August 2023	Charolais	Bull	0.9756	<0.0001
14 July 2023	Angus	Bull	0.9872	<0.0001
7 August 2023	Angus	Bull	0.9827	<0.0001
14 July 2023	Charolais	Heifer	0.9887	<0.0001
7 August 2023	Charolais	Heifer	0.9850	<0.0001
14 July 2023	Angus	Heifer	0.9887	<0.0001
7 August 2023	Angus	Heifer	0.9653	<0.0001

**Table 2 animals-15-02904-t002:** Dry matter intake (both measured and determined by the GPBoost method of [19] and water intake for each breed by class group.

Group	n	Average Dry Matter Intake, kg		Average Water Intake, L
		Measured	Determined	
Charolais bulls	35	10.75 ± 0.20	10.12 ± 0.11	41.4 ± 1.9
Angus bulls	57	10.75 ± 0.16	10.12 ± 0.09	41.5 ± 1.5
Charolais heifers	24	9.07 ± 0.26	9.79 ± 0.14	32.0 ± 2.4
Angus heifers	28	8.17 ± 0.24	9.37 ± 0.13	26.3 ± 2.2

## Data Availability

Data presented here may be available upon request to the corresponding author.

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
