# Peer review of "Technical Note: Assessment of a Novel Method to Measure Water Intake in Beef Cattle and Its Application to Determining Dry Matter Intake"

_animals, 2025, doi:10.3390/ani15192904_

Round 1
Reviewer 1 Report
Comments and Suggestions for Authors
General comments to the author
The technical note is generally well put together and succinctly explained. The methods are unique and timely, though calling in line water meters synced via timestamp “novel” is a bit of a stretch. There are some concerns regarding the validation methods used. In particular, measuring total water usage and comparing against the cumulative sum of water use per water meter only validates the accuracy of the individual water meters, not the accuracy and precision of assigning water usage to individual animals. Authors should review Tedeschi, 2006 (Assessment of the adequacy of mathematical models) for methods to calculate the precision and accuracy of systems and predictive models. Further, time is not a factorial variable, particularly when it represents measurements taken continuously over the study period. This is a critical statistical flaw that needs to be rectified. Also, I looked twice, but did not see how dry matter intake was predicted using water intake and compared to measured dry matter intake values. Here the authors should be careful to consider model calibration and validation procedures. Otherwise, the manuscript is generally well written, easy to follow, and a pertinent topic for the time using a relatively accessible method for research institutions. I look forward to seeing the next reveisions for this paper.
Specific comments:
Line 36: The statement says that in-pen weights were not different from chute weights, but presents with a highly significant p-value which would indicate that highly significant differences were found.
Line 37: This validates the accuracy of the in-pen water meters, not the accuracy and precision of applying water consumption to individual animals. Given the title and context of the paper, and the manner in which the data is presented, this represents a bit of sleight of hand that should be more clearly represented, as it would be confusing to future readers.
Line 144 – 47: Clarify how much missing data was imputed, and how the actual distribution was calculated. The quantity of missing data would affect the confidence of the imputed values, particularly if they were pulled from distributions representing each factorial subset (animal class, breed, time) or the distribution of the whole dataset, would make a significant difference in the level of confidence in the final results.
Line 180: Time is a continuous variable, While it is possible to sometimes use time as a factorial variable, this is only valid if there is sufficient temporal separation between consecutive data points. For data such as daily feed, water, and body-weight, this is most certainly not a valid choice. This is a fixable, but fatal flaw in the current methodologies that must be rectified.
Line 196: There are numerous methods to compare the accuracy and precision between two forms of measuring a variable. Please refer to Tedeschi, 2006 for methods to assess the accuracy of models, and Wells et al., 2021 and Parsons et al 2023 for examples of applying this methodology to in-pen and chute collected bodyweights.
Line 206: Use Cattle, not Cows, on the Y axis label.
Line 230 – 235: Same comment as above, this is only measuring the cumulative error rate of the individual pen meters, not the accuracy and precision of assigning individual water intake to animals. While not a fatal flaw, I don’t find this information informative to the objective of the paper, and worse could be misleading to readers who may interpret this as the water being directly assigned to the animal.
Line 251: How was determined dry matter intake calculated? Was this before missing data was imputed vs after, or is it a “missing” prediction equation using water intake to predict feed intake? If it is the former, please provide information regarding the overall quality and quantity of collected/missing data. If it is the latter, please provide the methodology used to develop the prediction equation, and report model training, validation metrics and statistics.
Line 307: Again, you have not demonstrated that this method is able to accurately assign individual water intake to individual animals in the same fashion that feed intake is assigned to individual animals. The data only validates the accuracy of the in pen flow meters, not the ability to assign that water to individual animals. Issues such as splash over, fill rate, etc all play a role that would not effect the difference between total system flow and the cumulative sum of water used at individual waterers.
Reviewer 2 Report
Comments and Suggestions for Authors
Very good and innovative work. It has interesting projections.
I suggest improving the presentation of the figures, especially numbers 1, 2, and 3.
I suggest discussing whether this methodology could be extended to cattle and sheep under extensive grazing conditions.
Reviewer 3 Report
Comments and Suggestions for Authors
The aim of this technical note is to evaluate the accuracy of some tools to measure weight and water intake. This work tried to verify Vytelle In-Pen Weighing Position validity and also to validate the use of a flow meter in line with a commercial waterer to measure daily water intake. The result of this work is very interesting in the context of climate changes and the water scarcity. The manuscript is well written, but the materials and methods section should be reviewed. It should be subdivided in subsections, and data analyses section should be clearly described. There are some comments presented below.
-In material and methods section, could you mention how many animals are in each pens.
-I recommend that authors present feed composition in table, it will be better.
- Material and method section should be divided in other section to be more organized, informative, and explicit.
- Please add a section of data analyses in material and method section.
- In line 210, I think the text is merged with the title of the figure.
- Please add the name of the method 16 , the titles of table 2 and figure 6.
- Please improve the title of figures 8 and 9, they should be more informative.
Round 2
Reviewer 1 Report
Comments and Suggestions for Authors
General comments:
The manuscript is generally improved and will be greatly strengthed with the inclusion of information provided in responses below into the main text. However, it is not accurate to describe this manuscript as a validation of the accuracy and precision of the water intake monitoring equipment to track individual animal water consumption. A validation requires comparison of a systems captured data to an alternative or “gold standard” source measured at the same resolution. Given that the title of the paper reads in part “Validation of a novel method to measure water intake”, the absence of this comparison constitutes a fatal flaw as the paper does not adequately take on the methods necessary to conduct such a validation.
To whom it may concern:
We thank the reviewers for their thoughtful input on our manuscript animals-3819573. We have
revised the manuscript according to their comments and give a detailed response to each
recommendation below. We have highlighted where we have made changes to the manuscript to
facilitate further review. We feel these changes improve the manuscript and look forward to its
publication.
Reviewer 1
The technical note is generally well put together and succinctly explained. The methods are unique and timely, though calling in line water meters synced via timestamp “novel” is a bit of a stretch. There are some concerns regarding the validation methods used. In particular, measuring total water usage and comparing against the cumulative sum of water use per water meter only validates the accuracy of the individual water meters, not the accuracy and precision of assigning water usage to individual animals. Authors should review Tedeschi, 2006 (Assessment of the adequacy of mathematical models) for methods to calculate the precision and accuracy of systems and predictive models. Further, time is not a factorial variable, particularly when it represents measurements taken continuously over the study period. This is a critical statistical flaw that needs to be rectified. Also, I looked twice, but did not see how dry matter intake was predicted using water intake and compared to measured dry matter intake values. Here the authors should be careful to consider model calibration and validation procedures. Otherwise, the manuscript is generally well written, easy to follow, and a pertinent topic for the time using a relatively accessible method for research institutions.
I look forward to seeing the next revisions for this paper.
Response: The reviewer’s general comments are helpful, and we address specifics below. We have expanded the description of the individual water intake collection and drawn parallels to the feed intake equipment from the same vendor.
Specific comments:
Reviewer 1: Line 36: The statement says that in-pen weights were not different from chute weights, but presents with a highly significant p-value which would indicate that highly significant differences were found. Response: We have clarified that the p-value is for Rho. Rho is a nonparametric correlation coefficient and this indicates high correlation and not differences (lines 36-40 and 229-230).
Reviewer 1 Response: this is much better thank you
Reviewer 1: Line 37: This validates the accuracy of the in-pen water meters, not the accuracy and precision of applying water consumption to individual animals. Given the title and context of the paper, and the manner in which the data is presented, this represents a bit of sleight of hand that should be more clearly represented, as it would be confusing to future readers.
Response: The dilemma is that there is no “gold standard” of water consumption by individual cattle. Tedeschi (2006) states, it is sometimes impossible to validate certain values in animal physiology, like precise number of microbes in the rumen. This is a similar scenario, when there is no objective and accurate measure of the animal’s drinking volume in undisturbed conditions, unless one surgically installed the water flow meter into animals’ esophagus. Even that would be grossly biased due to the ruminants’ – specific behavior of regurgitation of liquefied feed matter from the rumen back to the mouth and chewing and swallowing that again. We have added detail to the text to describe how the individual flow meter data is collected in an integrated way into the Vytelle data acquisition panel and the individual intakes are retrieved from their database (in the same way the body weight data is collected and retrieved; lines 129-132, 136-138 and 140-142).
Reviewer 1 Response: I understand the issue, but I don’t think such drastic measures are necessary. Please keep in mind the lag effect where the rate of water consumption may be greater than water inflow, which would create errors in the individual assignment of water consumed per drinking bout. Humor me the following scenario, where animal 1 enters the full waterer at time 1, and commences drinking until time 2, when he is replaced by animal 2.
W = Full water level
F = Inflow rate
T = Time
C = Water consumption rate
Here:
Measured consumption by Animal 1 = W – (c x (t2-t1)) + (F x (t2-t1))
And
Measured consumption by animal 2 = yt2 + (c2 x (t3-t2) + f x (t3-t2))
Assuming that:
F is relatively constant
C of animal 1 > F
C of animal 2 < F
Since C is unknown and likely highly variable both between animals and drinking bouts within animal, the potential lag in water inflow would generate error in the system if water is assigned if the difference in reservoir levels (yt2) is not accounted for.
Appeal to the difficulty of measuring a parameter, or the lack of a “gold standard” does not justify substitution of a different parameter unrelated to the original question, which is as I understand it, validation of systems to measure individual water intake in group housed cattle. Comparing the cumulative sum of individual water meters to the total water usage only validates the accuracy of the individual meters and the fact that there is not a water leak or another source of water disappearance between the residential meters and the flow meters. It does not in validate the accuracy or precision of allocating water to individual animals.
Reviewer 1: Line 144 – 47: Clarify how much missing data was imputed, and how the actual
distribution was calculated. The quantity of missing data would affect the confidence of the imputed values, particularly if they were pulled from distributions representing each factorial subset (animalclass, breed, time) or the distribution of the whole dataset, would make a significant difference in the level of confidence in the final results.
Response: To clarify, we did not impute missing values by sampling from statistical distributions.
Instead, we used a time-series– and similarity-based strategy (see [19] for more detail). In our dataset of 6,218 records, missing values were limited (dailyWaterVisits (201 records), timeAtWater (210 records), timePerWaterVisit (210 records), dailyWaterIntake (210 records) 3.4% each and bodyweight (52 records) 0.8%). Missingness was imputed using a stepwise approach: long gaps (>3 days) were interpolated (PCHIP for intake and water traits, linear for age), short gaps (≤3 days) were imputed using KNN (k=6), and edge cases were handled with forward/backward fill. This ensured that imputations preserved within-animal temporal structure rather than relying on overall dataset distributions, thereby minimizing potential bias.
Reviewer 1 response: This is a great answer. Please include in your methods and results.
Reviewer 1: Line 180: Time is a continuous variable, While it is possible to sometimes use time as a factorial variable, this is only valid if there is sufficient temporal separation between consecutive data points. For data such as daily feed, water, and body-weight, this is most certainly not a valid choice. This is a fixable, but fatal flaw in the current methodologies that must be rectified.
Response: We indicated in Materials and methods and in - Fig 2 “Distribution of chute (gold) and IPW (blue) weights (kg) in Angus and Charolais bulls and heifers on two occasions”, there were two (non-consecutive dates) dates, when both the chute weight and IPW were taken on the same time (lines 179-182). These dates were three weeks apart, so the time here could be considered as a factorial fixed factor, for which the LS-means could be generated by the model. More frequent (daily) chute weights could be stressful to animals and may have adverse effect on the weight gain for animals on a gain test. However, we accounted for the repeated nature of the time (and weighing method) in our mixed effects model with doubly-repeated measures; the animal was labelled as a subject for random factor (on which the date and weighting method were repeated) and fitting random intercept to each breed took care of possible correlation of measures within each breed. In addition, we ran the model using ‘Date’ in a repeated random row and in this model the p-value for the effect of method was 0.572. However, this approach would not allow us to evaluate the date x method interaction. Thus, we appeal to the statement that our methodology is flawed. However, we respectfully appreciate the reviewer’s suggestion, and we added the extra explanation in addition to adding a citations to our statistical methodology section. Here we provide the SAS code used:
PROC MIXED data=MW_BW_DOUBLYREPEATED;
Title "Summer2023_BW_IPW Upd with 2 days validataion_with chute_doubly repeated ANOVA";
Class Breed Class EID_MID_f_past84000 Method Date ;
Model VAR12 = Class|Breed|Date|Method / ddfm=kr;
Random intercept Breed / subject= EID_MID_f_past84000;
Repeated Method Date / type=un@ar(1) subject = EID_MID_f_past84000;
LSMeans Class Breed Method Date Breed*Method Method*Date Class*Method*Date/diff adjust=tukey ;
run;
quit;
where EID_MID_f_past84000 is a unique identifier of the animal ID.
Reveiwer 1 response: Thank you for your inclusion of the model. This answered my question.
Reviewer 1: Line 196: There are numerous methods to compare the accuracy and precision between two forms of measuring a variable. Please refer to Tedeschi, 2006 for methods to assess the accuracy of models, and Wells et al., 2021 and Parsons et al 2023 for examples of applying this methodology to in-pen and chute collected bodyweights.
Response: We used the RMSE = root mean square error (similar to MSE and MSEP) used in Tedeschi (2006) and Bland-Altman plot and correlation analyses that were used by Well et al (2021). In addition to testing the parallelism of the curves.
Reveiwer 1 response. I see where RMSE is used to compare water and feed consumption, but not body weight. Also, consider the issue addressed above that this experiment did not capture the data necessary to calculate RMSE on an individual animal basis for water consumption.
Reviewer Line 206: Use Cattle, not Cows, on the Y axis label.
Response: This has been corrected.
Reviewer 1: Line 230 – 235: Same comment as above, this is only measuring the cumulative error rate of the individual pen meters, not the accuracy and precision of assigning individual water intake to animals. While not a fatal flaw, I don’t find this information informative to the objective of the paper, and worse could be misleading to readers who may interpret this as the water being directly assigned to the animal.
Response: We have added additional description to the methods. The individual flows are collected in the same DAQ panel as the body weight data using the same RFID antenna and integrated into the Vytelle system. Their proprietary software assigns water flow to the animal that is present and bowl recharge for a discreet period after the animal leaves to that animal.
“The flow meter data was recorded via the Vytelle Data Acquisition Panel and thereby integrated
with the RFID recordings and load cell recordings at the In-Pen Weighing Position. The software for the intake system measures all flow through the meters and then as-sociates given flows to the RFID tag present in the In-Pen Weighing Position at that time. Proprietary components of the software assign flows (including immediately after an animal leaves the In-Pen Weighing Position) to the same animal to which the weight data is assigned” (lines 136-142)
Reveiwer 1 Response: This is helpful, and directly addresses the issue for the scenario I put forward in my response to line 37. However, since this is a validation of individual animal water intake and only one parameter of individual water intake was measured, it is not possible to validate that this algorithm is functioning correctly. Helpful statistics supporting validity of the water intake results would include a similar table to Table 2 describing feed intake. Please include the above description of the Vytelle algorithm in the methods section of the manuscript, as it is critical information to understanding the paper.
Reviewer 1: Line 251: How was determined dry matter intake calculated? Was this before missing data was imputed vs after, or is it a “missing” prediction equation using water intake to predict feed intake? If it is the former, please provide information regarding the overall quality and quantity of collected/missing data. If it is the latter, please provide the methodology used to develop the prediction equation, and report model training, validation metrics and statistics.
Response: The determined dry matter intake was done by inference using the GPBoost method of [19]. The specific method from that reference has been added to the text and the figure legends for clarity. This is our current use case for the daily body weight and water intake data and wanted to demonstrate a use for determining daily water intake data. The details of the methodology are found in [19] (lines 207-210, 289-290 and 301).
Reviewer 1 Response: Was determined dry matter intake calculated before or after imputing missing results. Thank you for the inclusion of additional clarification. This is a great use case for water intake.
Reviewer 1: Line 307: Again, you have not demonstrated that this method is able to accurately assign individual water intake to individual animals in the same fashion that feed intake is assigned to individual animals. The data only validates the accuracy of the in-pen flow meters, not the ability to assign that water to individual animals. Issues such as splash over, fill rate, etc all play a role that would not effect the difference between total system flow and the cumulative sum of water used at individual waterers.
Response: We disagree. Feed intake from bunks is assigned by determining a difference in bunk
weight and assigning that to the animal that is present. There is no way to assess in that case that the feed was actually consumed. Using the same data collection framework, here the flow data is
integrated into the data acquisition panel and is assigned to the animal that is present (including the flow to complete the bowl recharge immediately after an animal leaves). The flow monitoring is fully integrated into the Vytelle system, the data on individual water intakes retrieved from the same database where the body weight data is retrieved. The sum of all of those individual daily water intakes was then compared to the sum of all flow into the facility using a separate meter and a very high level of agreement was found over 60 days (157 animals x 60 days x an average of 5 visits/day ~ 47,000 intake records).
Reviewer 1 Response: Validation of the accuracy of individual animal feeding and drinking bouts requires direct observation of the bouts in monitored. Examples for feed intake include Chapinal et al 2007 with the insatec feed intake system or Oliveria et al. 2018 with the Intergado system and for water intake Allwardt et al 2017 with the insatec water intake system, all of which employed direct observers to monitor the disappearance or consumption of water on an individual bout basis to serve as their standard with which to compare the automatically captured intake data against for their validation methodologies. A note on papers published by Devries et al., 2003, Mendes et al., 2011, Bach et al., 2004, and Schwartzkoph-Genswein et al 1999 is that they validated feeding behavior, not feed intake, and used direct observers either via in person or camera systems to monitor animal feeding behavior.

Round 3
Reviewer 1 Report
Comments and Suggestions for Authors
The manuscript is greatly improved and I recommend this manuscript for publication.